REGISTERED REPORT

# Registered report: *Fusobacterium nucleatum* infection is prevalent in human colorectal carcinoma

**John Repass[1], Nimet Maherali[2], Kate Owen[3], Reproducibility Project: Cancer Biology\***

[1]ARQ Genetics, Bastrop, United States; [2]Harvard Stem Cell Institute, Cambridge, United States; [3]University of Virginia, Charlottesville, United States

## REPRODUCIBILITY
### PROJECT
## CANCER BIOLOGY

**Abstract** The Reproducibility Project: Cancer Biology seeks to address growing concerns about reproducibility in scientific research by conducting replications of selected experiments from a number of high-profile papers in the field of cancer biology. The papers, which were published between 2010 and 2012, were selected on the basis of citations and Altmetric scores (*Errington et al., 2014*). This Registered Report describes the proposed replication plan of key experiments from '*Fusobacterium nucleatum* infection is prevalent in human colorectal carcinoma' by Castellarin and colleagues published in *Genome Research* in 2012 (*Castellarin et al., 2012*). The experiment to be replicated is reported in Figure 2. Here, Castellarin and colleagues performed a metagenomic analysis of colorectal carcinoma (CRC) to identify potential associations between inflammatory microorganisms and gastrointestinal cancers. They conducted quantitative real-time PCR on genomic DNA isolated from tumor and matched normal biopsies from a patient cohort and found that the overall abundance of *Fusobacterium* was 415 times greater in CRC versus adjacent normal tissue. These results confirmed earlier studies and provide evidence for a link between tissue-associated bacteria and tumorigenesis. The Reproducibility Project: Cancer Biology is a collaboration between the Center for Open Science and Science Exchange and the results of the replications will be published in *eLife*.

**\*For correspondence:** nicole@ scienceexchange.com

**Group author details:**
Reproducibility Project: Cancer Biology See page 8

## Introduction

The human intestine is populated by an estimated $10^{14}$ microbes comprising over 1000 bacterial phylotypes (*Ley et al., 2006*). The overall composition of the intestinal microbiota is determined by a number of factors, including host genetics, environment, diet and hygiene (*Arrieta et al., 2014*; *Keku et al., 2015*). These bacteria play important roles in host biology by maintaining intestinal homeostasis, barrier function, immunity and metabolic function (*Backhed et al., 2005*; *Jones et al., 2014*). Perturbations or imbalances in the microbiome (microbial dysbiosis) are linked to a number of disease pathologies such as inflammatory bowel disease (*Collins, 2014; Hold et al., 2014*), obesity (*Bajzer and Seeley, 2006*; *Brown et al., 2012*), and colorectal cancers (CRCs; *Dulal and Keku, 2014*; *Keku et al., 2015*).

CRC is a complex disease arising from the sequential accumulation of somatic mutations and epigenetic alterations. Activating mutations in the *K-ras* oncogene, as well as the loss of tumor suppressor genes like *p53* (*TP53*) and adenomatous polyposis coli (*APC*), contribute to the tumorigenic transformation of normal colonic epithelium (*Vogelstein et al., 1988*; *Fearon, 2011*; *Mundade et al., 2014*). In addition to genetic factors, microbial dysbiosis, such as altered bacterial diversity, is strongly associated with the development of CRC (*Keku et al., 2015*). However, despite numerous longitudinal studies comparing intestinal microbial communities over time

(*Rodriguez et al., 2015*), and across various cancer stages (*Kubota, 1990; Chen et al., 2013; Nugent et al., 2014*), there is limited information on the contribution of specific bacteria to CRC development.

To identify potential associations between inflammatory microorganisms and gastrointestinal cancers, *Castellarin et al. (2012)* first performed RNA sequencing (RNA-seq) on a limited number of tumor and matched normal tissue samples. Initial observations indicated a striking overrepresentation of *Fusobacterium nucleatum* sequences in carcinoma samples compared to controls. To confirm these findings, *Castellarin et al. (2012)* assessed the relative abundance of *Fusobacterium* in a larger cohort of tumor and matched normal biopsy samples. In Figure 2, the authors performed quantitative real-time PCR (qPCR) on genomic DNA (gDNA) isolated from an additional 88 colorectal carcinoma (CRC) specimens and adjacent matched control tissues. *Fusobacterium* abundance was observed to be significantly higher in the tumor samples compared to matching control samples. This key experiment will be replicated in Protocol 1.

Similar findings confirming the higher relative abundance of *Fusobacterium* in CRC tumor tissues compared to control biopsies have been reported by other investigators (*Kostic et al., 2012*; *McCoy et al., 2013*; *Warren et al., 2013*; *Tahara et al., 2014*). In fact, the study by *Kostic et al. (2012)* is considered a co-discovery of this phenomenon. *McCoy et al. (2013)* successfully validated the association between *Fusobacterium* and CRC in a set of matched CRC tumor and normal human colon tissue samples using both pyrosequencing and qPCR analysis of the *16S* bacterial rRNA gene. Findings by *Mira-Pascual et al. (2015)* further confirm this trend, as this group observed a significantly higher presence of *F. nucleatum* in mucosal samples from the CRC patients compared to the healthy subjects (as opposed to matched tissue biopsies). Recent studies have also reported a higher presence of *Fusobacterium* species in human colonic adenomas (polyps) and in stool samples from adenoma and tumor carcinoma patients compared to healthy subjects (*Kostic et al., 2012*; *2013*; *McCoy et al., 2013*). Furthermore, other studies have expanded these findings to identify potential mechanisms of action of *F. nucleatum* during tumorigenesis (*Rubinstein et al., 2013*; *Gur et al., 2015*). *Rubenstein et al. (2013)* also indirectly confirm a higher abundance of *Fusobacterium* in CRC patients by measuring higher *F. nucleatum* FadA mRNA expression relative to healthy controls.

## Materials and methods

Unless otherwise noted, all protocol information was derived from the original paper, references from the original paper, or information obtained directly from the authors. An asterisk (*) indicates data or information provided by the Reproducibility Project: Cancer Biology core team. A hashtag (#) indicates information provided by the replicating lab.

### Protocol 1: quantitative PCR for amplification of *F. nucleatum* from matched normal and tumor human colon cancer specimens

This protocol utilizes quantitative PCR to test the relative abundance of *F. nucleatum* DNA in gDNA isolated from matched normal and tumor human colon cancer specimens. It is a replication of Figure 2.

### Sampling

- This experiment will include 40 matched samples for a final power of 87.26%.
  - See power calculations for details.
- Each patient sample has two cohorts:
  - Cohort 1: Colon tumor sample (*n* = 40)
  - Cohort 2: Matched normal tissue within the same individual (*n* = 40)
  - Cohort 3: Age/ethnicity-matched normal tissue from additional control individuals (*n* = 40)
- Tissue is collected during surgery (either partial colectomy, ileocolectomy, colorectal resection, or proctocolectomy) from tumor tissue, adjacent normal tissue, or from normal controls. Samples are frozen on liquid nitrogen within 30 min after extractions. Diagnosis is confirmed by a pathologist using histological sections from each sample.
- Quantitative PCR will be performed for each sample two independent times in technical triplicate for the following:

- *F. nucleatum* DNA
- Prostaglandin transporter—reference gene

## Materials and reagents

| Reagent | Manufacturer | Catalog # | Comments |
|---|---|---|---|
| Frozen human colon tumor samples and matched normal samples | #iSpecimen | | Data include age, gender, ethnicity, diagnosis, histopathology report |
| Gentra Puregene Genomic DNA extraction kit | Qiagen | 158667 | Replaces Qiagen 69504 |
| PicoGreen Assay | #Life Technologies | P7589 | |
| Spectrophotometer | #NanoDrop | ND1000 | |
| 384-well optical PCR plate | #Phoenix Research | MPS-3898 | |
| Fusobacteria forward qPCR primer | Part of a custom-designed Taqman primer/probe set (Applied Biosystems) | | CAACCATTACTTTAACTCTACCATGTTCA |
| Fusobacteria reverse qPCR primer | | | GTTGACTTTACAGAAGGAGATTATGTAAAAATC |
| Fusobacteria FAM probe | | | TCAGCAACTTGTCCTTCTTGATCTTTAAATGAACC† |
| PGT forward qPCR primer | Part of a custom-designed Taqman primer/probe set (Applied Biosystems) | | ATCCCCAAAGCACCTGGTTT |
| PGT reverse qPCR primer | | | AGAGGCCAAGATAGTCCTGGTAA |
| PGT FAM probe | | | CCATCCATGTCCTCATCTC |
| TaqMan Universal Master Mix | ABI | #4304437 | |
| qPCR thermal cycling system | ABI | #4351405 | 7900HT system |

†Note: Probe sequence from original manuscript incorrect. Correct sequence seen here from *Flanagan et al., 2014*.

## Procedure

1. Obtain ~40 sets from frozen human CRC tumors with matched normal control, and an additional control group of age/ethnicity-matched tissue from healthy individuals.
   a. Tissue will have been flash-frozen in liquid nitrogen very soon after harvest.
   b. Pathological data showing positive diagnosis for CRC will be included with samples.
2. Extract gDNA using Gentra Puregene genomic DNA extraction kit according to manufacturer's instructions.
3. Quantify gDNA concentration by Nanodrop spectrophotometer.
4. Assemble 20 $\mu$L qPCR reactions in a 384-well optical PCR plate. Each sample is assayed in triplicate for each primer/probe set. Each reaction contains:
   a. 5 ng of gDNA
   b. 18 $\mu$M of each primer
   c. 5 $\mu$M of probe
   d. 1 X final concentration of TaqMan Universal Master Mix
5. Perform amplification and detection of DNA using the following reaction conditions:
   a. 2 min at 50°C
   b. 10 min at 95°C
   c. 40 cycles of 15 s at 95°C and 1 min at 60°C.
6. Calculate cycle threshold using the automated settings. Analyze and compute $\Delta\Delta C_T$ values by normalizing to prostaglandin transporter reference gene.
   a. The mean $\Delta\Delta C_T$ values from the technical replicates from the tumor and normal sample will be used to calculate the ratio of tumor versus normal for each matched biopsy.
7. Repeat steps 3–5 for each sample a second time.

a. The mean ratios of $\Delta\Delta C_T$ values in tumor versus normal sample from the two independent experimental replicates will be calculated for each matched biopsy.

## Deliverables

- Data to be collected:
  - Descriptive data of gDNA samples including: patient sample age/sex, ethnicity, and % area of the tumor involved with necrosis.
  - Purity ($A_{260/280}$ and $A_{260/230}$ ratios) and concentration of isolated total gDNA from tumor biopsies.
  - Raw qRT-PCR values, as well as analyzed $\Delta\Delta C_T$ values for each tumor and matched biopsy sample. Bar graph of mean relative abundance of *F. nucleatum* in tumor versus normal colorectal samples (compare to Figure 2A).

## Confirmatory analysis plan

This replication attempt will perform the statistical analysis listed below:

- Statistical analysis of replication data:
  - Note: At the time of analysis, we will perform the Shapiro–Wilk test and generate a quantile–quantile ($q$–$q$) plot to assess the normality of the data. If the data appear skewed, we will perform the appropriate transformation in order to proceed with the proposed statistical analysis. If this is not possible, we will perform the equivalent nonparametric test (e. g., Wilcoxon-signed rank test).
  - One-sample Student's *t*-test using the log of the mean ratios of $\Delta\Delta C_T$ values from the two independent experimental replicates, tumor $\Delta\Delta C_T$/matched within individual controls compared to a mean value of zero.
- Additional exploratory analysis:
  - Two Student's *t*-tests with Bonferroni correction comparing absolute values from:
    - Mean tumor *Fusobacterium* abundance versus within subject matched control (paired)
    - Mean tumor *Fusobacterium* abundance versus healthy matched control (unpaired)
- Meta-analysis of original and replication attempt effect sizes:
  - Compute the effect size, compare it against the effect size in the original paper and use a random effects meta-analytic approach to combine the original and replication effects, which will be presented as a forest plot.

## Known differences from the original study

All known differences are listed in the 'Materials and reagents' section with the originally used item listed in the comments section. All differences have the same capabilities as the original and are not expected to alter the experimental design. We have added an additional control of matched gDNA from healthy individuals.

## Provisions for quality control

The sample purity ($A_{260/280}$ and $A_{260/230}$ ratios) of the isolated gDNA from each sample will be reported. All of the raw data, including the analysis files, will be uploaded to the project page on the OSF (https://osf.io/v4se2) and made publically available.

## Power calculations

For a detailed breakdown of all power calculations, see spreadsheet at https://osf.io/yadgq/

## Protocol 1

### Summary of original data

- Note: Data estimated from graph reported in Figure 2.

| Sample | Log (mean) | N |
|---|---|---|
| 1 | 1.5787 | 2 |
| 2 | 1.1957 | 2 |
| 3 | 0.9277 | 2 |
| 4 | 0.8766 | 2 |
| 5 | 0.5192 | 2 |
| 6 | 0.4468 | 2 |
| 7 | 0.4128 | 2 |
| 8 | 0.3149 | 2 |
| 9 | 0.2936 | 2 |
| 10 | 0.2681 | 2 |
| 11 | 0.2766 | 2 |
| 12 | 0.2383 | 2 |
| 13 | 0.234 | 2 |
| 14 | 0.2 | 2 |
| 15 | 0.1787 | 2 |
| 16 | 0.1703 | 2 |
| 17 | 0.1617 | 2 |
| 18 | 0.1362 | 2 |
| 19 | 0.0681 | 2 |
| 20 | 0.0298 | 2 |
| 21 | 0.034 | 2 |
| 22 | 0.0128 | 2 |
| 23 | 0.0095 | 2 |
| 24 | 0.017 | 2 |
| 25 | 0.0213 | 2 |
| 26 | 0.0213 | 2 |
| 27 | 0.0255 | 2 |
| 28 | 0.0128 | 2 |
| 29 | 0.017 | 2 |
| 30 | 0.0128 | 2 |
| 31 | 0.017 | 2 |
| 32 | 0.0255 | 2 |
| 33 | 0.0213 | 2 |
| 34 | 0.0301 | 2 |
| 35 | 0.034 | 2 |
| 36 | 0.0555 | 2 |
| 37 | 0.1362 | 2 |
| 38 | 0.1447 | 2 |
| 39 | 0.1745 | 2 |
| 40 | 0.1915 | 2 |
| 41 | 0.2 | 2 |
| 42 | 0.2086 | 2 |

*Continued*

| Sample | Log (mean) | N |
|---|---|---|
| 43 | 0.217 | 2 |
| 44 | 0.2213 | 2 |
| 45 | 0.2596 | 2 |
| 46 | 0.4043 | 2 |
| 47 | 0.4468 | 2 |
| 48 | 0.4511 | 2 |
| 49 | 0.4681 | 2 |
| 50 | 0.4979 | 2 |
| 51 | 0.5064 | 2 |
| 52 | 0.5021 | 2 |
| 53 | 0.549 | 2 |
| 54 | 0.5787 | 2 |
| 55 | 0.5787 | 2 |
| 56 | 0.5872 | 2 |
| 57 | 0.6085 | 2 |
| 58 | 0.6213 | 2 |
| 59 | 0.6553 | 2 |
| 60 | 0.6979 | 2 |
| 61 | 0.7234 | 2 |
| 62 | 0.7617 | 2 |
| 63 | 0.8043 | 2 |
| 64 | 0.8298 | 2 |
| 65 | 0.966 | 2 |
| 66 | 0.9617 | 2 |
| 67 | 1.0042 | 2 |
| 68 | 1.0128 | 2 |
| 69 | 1.017 | 2 |
| 70 | 1.0255 | 2 |
| 71 | 1.0681 | 2 |
| 72 | 1.0596 | 2 |
| 73 | 1.0851 | 2 |
| 74 | 1.1234 | 2 |
| 75 | 1.1958 | 2 |
| 76 | 1.3149 | 2 |
| 77 | 1.3149 | 2 |
| 78 | 1.4085 | 2 |
| 79 | 1.6298 | 2 |
| 80 | 1.7575 | 2 |
| 81 | 1.783 | 2 |
| 82 | 1.8723 | 2 |
| 83 | 1.9404 | 2 |
| 84 | 1.983 | 2 |

*Continued on next page*

*Continued*

| Sample | Log (mean) | N |
|---|---|---|
| 85 | 2 | 2 |
| 86 | 2.2553 | 2 |
| 87 | 2.4298 | 2 |
| 88 | 2.4723 | 2 |
| 89 | 2.4723 | 2 |
| 90 | 2.5532 | 2 |
| 91 | 2.6723 | 2 |
| 92 | 2.6893 | 2 |
| 93 | 2.9064 | 2 |
| 94 | 3.0596 | 2 |
| 95 | 3.2425 | 2 |
| 96 | 3.3447 | 2 |
| 97 | 3.5872 | 2 |
| 98 | 3.8 | 2 |
| 99 | 4.261 | 2 |

## Test family

- Ratio one-sample *t*-test: $a_{error}$ = 0.05, μ = 0.

## Power calculations

- Ratio *t*-test and power calculations were performed with R software, version 3.1.2 (***Team RC 2014***).

| | Mean | Effect size *d* | A priori power | Total sample size |
|---|---|---|---|---|
| Ratio | 0.75893838 | 0.5024568 | 87.26% | 40* |

*Forty total ratios (40 tumor 40 matched controls) will be used.

## Additional exploratory analysis
### Test family

- Paired Student's *t*-test (two-tailed): $a_{error}$ = 0.025.

### Power calculations

- Sensitivity calculations were performed with G*Power software, version 3.1.7 (***Faul et al., 2007***).

| Group 1 | Group 2 | Detectable effect size *d* | A priori power | Total sample size |
|---|---|---|---|---|
| Tumor sample | Adjacent matched control | 0.50384 | 80% | 40 |

## Test family

- Independent Student's *t*-test (two-tailed): $a_{error} = 0.025$.

## Power calculations

- Sensitivity calculations were performed with G*Power software, version 3.1.7. (*Faul et al., 2007*).

| Group 1 | Group 2 | Detectable effect size *d* | A priori power | Total sample size |
|---------|---------|---------------------------|----------------|-------------------|
| Tumor sample | Healthy individual matched control | 0.7007 | 80% | 40 |

## Acknowledgements

The Reproducibility Project: Cancer Biology core team thank Courtney Soderberg at the Center for Open Science for assistance with statistical analyses. We also thank the following companies for generously donating reagents to the Reproducibility Project: Cancer Biology; American Type Culture Collection (ATCC), Applied Biological Materials, BioLegend, Charles River Laboratories, Corning, DDC Medical, EMD Millipore, Harlan Laboratories, LI-COR Biosciences, Mirus Bio, Novus Biologicals, Sigma–Aldrich, and System Biosciences (SBI).

## Additional information

### Group author details

Reproducibility Project: Cancer Biology

Elizabeth Iorns: Science Exchange, Palo Alto, United States; William Gunn: Mendeley, London, United Kingdom; Fraser Tan: Science Exchange, Palo Alto, United States; Joelle Lomax: Science Exchange, Palo Alto, United States; Nicole Perfito: Science Exchange, Palo Alto, United States; Timothy Errington: Center for Open Science, Charlottesville, United States

### Competing interests

JR: ARQ Genetics is a Science Exchange-associated lab. RP:CB: EI, FT, JL, NP: Employed by and hold shares in Science Exchange Inc. The other authors declare that no competing interests exist.

### Funding

| Funder | Author |
|--------|--------|
| Laura and John Arnold Foundation | Reproducibility Project: Cancer Biology |

The Reproducibility Project: Cancer Biology is funded by the Laura and John Arnold Foundation, provided to the Center for Open Science in collaboration with Science Exchange. The funder had no role in study design or the decision to submit the work for publication.

### Author contributions

JR, NM, KO, Drafting or revising the article; RP:CB, Conception and design; Drafting or revising the article

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
