## [Decision Letter]

Thank you for submitting your work entitled "Registered report: *Fusobacterium nucleatum* infection is prevalent in human colorectal carcinoma" for consideration by *eLife*. Your article has been reviewed by four peer reviewers, and the evaluation has been overseen by a guest Reviewing Editor and Sean Morrison as the Senior Editor.

The reviewers have discussed the reviews with one another and the Reviewing Editor has drafted this decision to help you prepare a revised submission.

Summary:

The project goal is to test the reproducibility of the results in the Castellarin et al., 2012 paper in which *Fusobacterium* was associated with CRC by qPCR detection of *Fusobacterium* (versus a reference gene) in DNA isolated from colon tumors compared to matched uninvolved tissue and normal tissue from age-matched controls.

Essential revisions:

After discussion among all reviewers, the key clarifications requested are to define in more detail the source of the populations to be studied. Information about how subjects and controls will be recruited, how disease will be defined (including histopathology), how samples will be obtained as well as length and type of storage of samples should be explicitly stated. How many healthy individuals will be included in the study? Are tissues from healthy controls obtained through biopsy or colorectal resection? The title page should be corrected to reflect that the project is human subjects research.

[Editors’ note: a previous version of this study was rejected after peer review, but the authors submitted for reconsideration. The previous decision letter after peer review is shown below.]

Thank you for submitting your work entitled "Registered report: *Fusobacterium nucleatum* infection is prevalent in human colorectal carcinoma" for consideration at *eLife*. Your Registered Report has been evaluated by Sean Morrison (Senior editor) and a guest Reviewing Editor with appropriate expertise, and the decision was reached after discussions between the two of us. We regret to inform you that this version of the Registered Report will not be considered further for publication in *eLife*.

Unfortunately the proposed replication misses at least one key issue in this area of work. There are several reports of finding excess *Fusobacterium* in a subset of colon cancers versus matched normal and we have two major concerns:

1) Normal tissues from cancer hosts are not normal controls for these experiments (the authors would need to identify a bank of appropriate colonoscopy control biopsies or age-match normal tissues from rapid autopsies).

2) The key result that needs to be replicated is whether *Fusobacterium nucleatum* versus other species induce tumorigenesis in tumor-susceptible murine models. Addressing this key aspect of *Fusobacterium* biology by the Reproducibility Project is essential before we can move forwards with further in-depth review. Absent this experiment, it will not be possible to conclude whether they key claims in the paper in question are reproducible.

---

## [Author Response]

Essential revisions:

After discussion among all reviewers, the key clarifications requested are to define in more detail the source of the populations to be studied. Information about how subjects and controls will be recruited, how disease will be defined (including histopathology), how samples will be obtained as well as length and type of storage of samples should be explicitly stated. How many healthy individuals will be included in the study? Are tissues from healthy controls obtained through biopsy or colorectal resection? The title page should be corrected to reflect that the project is human subjects’ research.

We have added clarifications to the manuscript to address these questions, but have answered more specifically below:

1) We will obtain tumor tissue samples from, iSpecimen located in Boston, MA. They have provided us with the information you requested.

2) The source population is defined as in the original manuscript: 50 years of age ( /- 5-10 years), normal control donors are matched for age, gender, and ethnicity and are collected from patients during surgery (either partial colectomy, ileocolectomy, colorectal resection or proctocolectomy). Samples are frozen in liquid nitrogen within 30 minutes of extraction.

3) iSpecimen represents multiple medical centers which use in-house pathologists to conduct histology on sections of all samples to confirm diagnosis.

4) The kit for gDNA extraction is a similar kit from Qiagen and will quantify gDNA using a NanoDrop.

Since the Registered Report does not involve using human tissue, but rather the proposed plan to use it, we typically do not list the Registered Report as a Human Subject study. However, we will include human subjects research information with the Final Report of replication results.

[Editors’ note: the author responses to the previous round of peer review follow.]

Unfortunately the proposed replication misses at least one key issue in this area of work. There are several reports of finding excess Fusobacterium in a subset of colon cancers versus matched normal and we have two major concerns:

1) Normal tissues from cancer hosts are not normal controls for these experiments (the authors would need to identify a bank of appropriate colonoscopy control biopsies or age-match normal tissues from rapid autopsies).

Adding additional aspects not included in the original study can be of scientific interest, and can be included if it is possible to balance them with the main aim of this project: to perform a direct replication of the original experiment(s). As such, we agree with the editors that there is scientific interest in including these additional control samples. We have added this control to the manuscript and made appropriate adjustments to our power calculations.

2) The key result that needs to be replicated is whether Fusobacterium nucleatum versus other species induce tumorigenesis in tumor-susceptible murine models. Addressing this key aspect of Fusobacterium biology by the Reproducibility Project is essential before we can move forwards with further in-depth review. Absent this experiment, it will not be possible to conclude whether they key claims in the paper in question are reproducible.

We agree that investigating whether this *Fusobacterium* ssp. in particular is responsible for inducing tumorigenesis is a critical extension of the Castellarin study. However, this project focuses on direct replication of the experiments as detailed in the original report and with information provided by the original authors to understand the reproducibility of the reported results. Aspects of an experiment not included in the original study are occasionally added (such as negative controls, reagent verification, etc.) to ensure the quality of the research, but are only included if it is possible to balance them with the main aim of this project: to perform a direct replication of the original experiment(s). We feel that the addition of this experiment, despite it being an important question in the field, is outside of the scope of this project.